# Diverging Flows: Detecting Out-of-Distribution Inputs in Conditional Generation

## Abstract

Flow matching models are able to learn complex conditional distributions from data. Nevertheless, they do not model the distribution of the conditioning itself, which means they can confidently generate samples from conditioning inputs that are not in the training distribution. In this work, we introduce *Diverging Flows*, an approach to train flow matching models that enables a single model to detect OOD conditions, without hindering its generative capabilities. *Diverging Flows* augments standard flow matching training with a contrastive objective that learns to separate the velocity fields produced by in- and out-of-distribution conditions, effectively modeling the conditions' distribution, and practically enforcing an effective telltale sign during the generation process. At inference time, we combine this signal with conformal prediction to obtain statistically valid OOD decisions. Additionally, *Diverging Flows* does not require real OOD data, enabling fully self-contained training on the target domain. The results indicate that *Diverging Flows* is competitive with other OOD detection methods while preserving the predictive quality of the underlying flow model. Ultimately, these results pave the way in adopting generative models as safe and robust predictors in high-stakes domains like weather forecasting, robotics, and medical applications.

## 1 Introduction

Many of the recent advances in machine learning are driven by diffusion models (Sohl-Dickstein et al., 2015; Ho et al., 2020), which enable the modeling of conditional multimodal data distributions and allow sampling from them. These models form the basis of state-of-the-art methods in image generation (Song et al., 2021; Karras et al., 2022) and imitation learning in robotics (Chi et al., 2023; Huang et al., 2024). Flow matching (FM) is a related approach that builds on ordinary differential equations and optimal transport (Lipman et al., 2022; Tong et al., 2023). In many cases, it is faster and easier to train than diffusion models, and is therefore increasingly used as a replacement in several domains (Jing et al., 2023; Jin et al., 2025; Chen & Lipman, 2024; Rouxel et al., 2025). In this work, we focus on flow matching, but extensions to diffusion remain possible.

By design, both diffusion and flow matching models capture aleatoric uncertainty, that is, the variability inherent in the data, and repeated sampling produces diverse outputs that reflect this data variability. However, they lack a built-in way to quantify epistemic uncertainty, i.e., uncertainty about the model itself and whether its predictions can be trusted. While epistemic uncertainty quantification is not critical in creative domains such as generating images or music (Mariani et al., 2024), it becomes essential in safety-critical settings like weather forecasting (Price et al., 2025) or robotic control (Rouxel et al., 2024), for which overconfident but incorrect predictions can lead to severe risks.

Epistemic uncertainty in conditional generation arises mainly from two sources: (1) the limited capacity of the model to represent the data (e.g., missing fine details, multimodality, not enough training, etc.) and (2) out-of-distribution (OOD) conditioning, which forces the model to extrapolate. In this work, we assume that modern models have a large capacity and that enough data is available to learn the distribution. We therefore focus on detecting out-of-distribution conditions, which requires upgrading the flow matching framework.

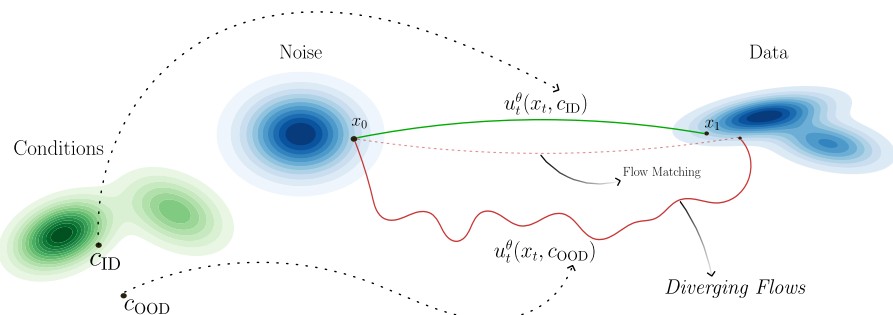

Figure 1: Conceptual overview of Diverging Flows. The model augments flow matching with a contrastive objective, forcing velocity fields induced by in-distribution and out-of-distribution conditions to diverge. At inference, the variability of the predicted velocities provides a natural signal for OOD detection.

A common strategy for OOD detection is to rely on explicit OOD datasets and cast OOD detection as classification problem. However, such data is rarely available in practice. One-class classifiers (Cui et al., 2023; Chalapathy & Chawla, 2019) offer an alternative, but they require training a second model, separate from the main generative model, with its own hyperparameters and design choices and computational overhead, complicating the overall framework. Our key intuition is that a distribution modeling process should, by itself, provide evidence when queried with inputs that lie outside its training distribution.

In this work, we exploit the underlying dynamics of the flow to extend the flow matching training loop, enabling the model to recognize when a conditioning input is out-of-distribution. Our main insight is that by contrasting the velocity fields induced by in- and out-of-distribution conditions, the model can implicitly learn the distribution of valid conditions without requiring explicit OOD data.

We accomplish this by incorporating contrastive learning into flow matching, enriching the model with a telltale signal that indicates whether a condition is OOD, without hindering its performance. Importantly, our approach does not rely on real OOD datasets: it learns to discriminate between in- and out-of-distribution conditions using only pseudo-OOD samples, such as Gaussian noise or adversarial perturbations. To further enable deployment in safety-critical settings, we combine this signal with conformal prediction, yielding statistically valid real-time OOD detection. While our method is motivated by the development of safe generative predictors, it can also be used as a standalone OOD detection module by performing reconstruction instead of prediction. We validate our approach through (i) a 2D toy example that illustrates how contrastive flow matching separates in- and out-of-distribution conditions, (ii) benchmark OOD detection tasks across CIFAR-10, SVHN, MNIST, and related datasets, and (iii) a real-world weather forecasting application where Diverging Flows achieves accurate prediction and near-perfect OOD detection under conformal calibration.

## 2 PRELIMINARY

### 2.1 FLOW MATCHING

Let us define $N$ $d$-dimensional data points $X = \{x^{(1)}, \ldots, x^{(N)}\} \in \mathbb{R}^d$, a source distribution $p$ (typically a standard Gaussian), and a target distribution $q$, where $X \sim q$. The goal of flow models is to transform samples $x_0 \sim p$ into samples $x_1 \sim q$. A flow $\psi_t : [0,1] \times \mathbb{R}^d \to \mathbb{R}^d$ evolves according to the ODE, $\frac{d}{dt}\psi_t(x_0) = u_t(\psi_t(x_0))$, with initial condition $\psi_0(x_0) = x_0$, where $u_t$ is a time-dependent velocity field. In practice, a flow model parameterizes this velocity field as a neural network

$$\frac{d}{dt}x_t = u_t^\theta(x_t). \tag{1}$$

Training by direct ODE simulation is costly. Flow matching avoids this by minimizing the discrepancy between predicted and target instantaneous velocities (Lipman et al., 2022):

$$\mathcal{L}_{\text{FM}} = \mathbb{E}_{t \sim \mathcal{U}[0,1],\, x_t}\left[\|u_t^\theta(x_t) - u_t(x_t)\|^2\right]. \tag{2}$$

To define $(x_t, u_t)$, we interpolate linearly between source and target: $x_t = (1 - t)x_0 + tx_1$ and $u_t = \dot{x}_t = x_1 - x_0$. After training, samples from $q$ can be generated by forward integrating the learned flow over a fixed number of steps.

## 2.2 Contrastive Learning

Contrastive learning aims to map data into a representation space where semantically similar inputs are close while dissimilar ones are far apart (Oord et al., 2018). Given a dataset $\{x^{(i)}\}_{i=1}^N$, we construct pairs of examples. For each $x^{(i)}$, we perform two random augmentations $t_1$ and $t_2$, producing a positive pair $x_{P1}^{(i)} = t_1(x^{(i)})$ and $x_{P2}^{(i)} = t_2(x^{(i)})$, which represent different views of the same sample. All other samples in the batch act as negatives relative to $x^{(i)}$.

An encoder $f_\theta : \mathbb{R}^d \to \mathbb{R}^k$ maps these inputs into a representation space, producing embeddings $z_{P1}^{(i)} = f_\theta(x_{P1}^{(i)})$, $z_{P2}^{(i)} = f_\theta(x_{P2}^{(i)})$, and negatives $z_N^{(j)} = f_\theta(x_N^{(j)})$. The training objective then encourages $z_{P1}^{(i)}$ and $z_{P2}^{(i)}$ to be close, while pushing them away from the negatives.

After training, the encoder $f_\theta$ produces representations that capture the semantic structure of the data and can be used for downstream tasks such as clustering, retrieval, or classification. Several loss formulations (e.g. InfoNCE (Oord et al., 2018; Chen et al., 2020), margin-based (Hadsell et al., 2006)) implement this idea, but they all share the same principle of bringing positives together and separating negatives.

## 3 Diverging Flows

Inspired by the contrastive learning paradigm, we propose *Diverging Flows*[1], a *contrastive flow matching* approach that incorporates OOD detection into conditional generation without hindering the model's performance. The key idea is to force flows under out-of-distribution conditions to diverge from the in-distribution ones, so that this divergence provides a reliable signal for OOD detection (Fig. 1). A flow model is practically a sampler that can sample data from a complex distribution $q(x_1)$. Conditional generation guides this process toward the conditional distribution $q(x_1 \mid c)$, where $c \in \mathbb{R}^d$ is the condition.

Following the formulation of guided probability paths (Lipman et al., 2024), we define

$$p_{t|c}(x_t \mid c) = \int p_{t|1}(x_t \mid x_1) \, q(x_1 \mid c) \, dx_1, \tag{3}$$

which interpolates between the source distribution at $t = 0$ and the conditional target distribution at $t = 1$. Importantly, this does not alter the mechanics of flow matching itself: the condition affects only the endpoint distribution $q(x_1 \mid c)$, while the interpolation path remains unchanged because

$$p_{t,1|c}(x_t, x_1 \mid c) = p_{t|1}(x_t \mid x_1) \, q(x_1 \mid c). \tag{4}$$

In many applications, conditions correspond to simple class labels. However, in predictive tasks, such as forecasting or trajectory prediction, the conditioning $c$ is often a high-dimensional modality (e.g., an image or state trajectory). Our focus in this work is on this latter, more challenging setting.

### 3.1 Contrastive Flow Matching

Our contrastive flow matching training does not follow directly the contrastive learning paradigm. In contrast to traditional contrastive learning, which operates on embeddings, our objective is to distinguish between flows rather than individual embeddings. To achieve that, let us first define $\tilde{c} \in \mathbb{R}^d$, which plays the role of an OOD condition, and $\psi_t$ and $\tilde{\psi}_t$, which are the in-distribution (ID) and OOD flows, respectively. Previously, we referred to the neural network $u_t^\theta$, which approximates the desired velocity field at time step $t$. Since, our work is based on the conditional generation case, we slightly change the definition of $u^\theta$, which now implements $u^\theta : (x_t, t, c) \mapsto u_t^\theta(x_t, c)$. Hence, during the contrastive training, we treat all predicted velocities $\hat{u}_{t,c}$ as the positive pairs and the $\hat{u}_{t,\tilde{c}}$ as the negative pairs.

---

[1]The code will be published upon acceptance

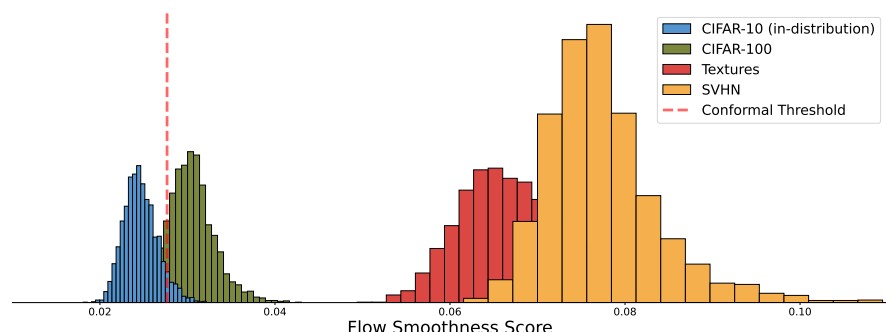

Figure 2: Histogram of the Flow Smoothness Score. We train our model on the CIFAR-10 dataset and perform out-of-distribution detection on SVHN, Textures and CIFAR-100

For making $\psi_t$ and $\tilde{\psi}_t$ diverge, we contrast $\hat{u}_{t,c}$ and $\hat{u}_{t,\tilde{c}}$. We accomplish that by adding two margin-based components (Balntas et al., 2016), $L_{\text{repel}}$ and $L_{\text{curve}}$, to the original flow matching:

$$L_{\text{repel}} = \max\{\|u_t - u_t^\theta(x_t, c)\|^2 - \|u_t - u_t^\theta(x_t, \tilde{c})\|^2 + \alpha, 0\} \qquad (5)$$

$$L_{\text{curve}} = \max\{(1 - \text{sim}(u_t, u_t^\theta(x_t, c))) - (1 - \text{sim}(u_t, u_t^\theta(x_t, \tilde{c}))) + \beta, 0\} \qquad (6)$$

where sim is the cosine similarity, $\alpha$ and $\beta$ are the margins, which represents the desired distance and angle between $\hat{u}_{t,c}$ and $\hat{u}_{t,\tilde{c}}$. At last, our loss function is shaped as following

$$L_{\text{DF}} = \|u_t - u_t^\theta(x_t, c)\|^2 + \lambda\, L_{\text{repel}} + \gamma\, L_{\text{curve}} \qquad (7)$$

with $\lambda$ and $\gamma$ being the weights of each component in the contrastive objective.

This new training objective equips the model with an implicit ability to capture the distribution of valid conditions. The repulsion term encourages flows under OOD conditions to deviate, while the curvature term further steers them away from the optimal trajectory. As a result, solving Eq. 1 yields flows that diverge substantially depending on the conditioning input. This divergence serves as a natural signal to decide whether a condition is in- or out-of-distribution, and thus whether the model's predictions can be trusted.

So far, we have described the training objective, but a crucial component remains: the construction of pseudo-OOD conditions. Our goal is not to train the model to discriminate between two datasets, but rather to calibrate its expectations about which types of conditions are valid. To this end, we generate artificial OOD conditions that allow the model to establish a separation boundary in the velocity field space. We explore two strategies:

1. **Gaussian noise**: We sample pseudo-conditions as $\tilde{c} \sim \mathcal{N}(0, I)$. Such inputs carry no structured information, encouraging the model to recognize that valid conditions should provide structured, feature-specific guidance to the generation process.

2. **Fast Gradient Sign Method (FGSM)**: We construct adversarial pseudo-conditions by perturbing a valid condition $c$ in the direction of the gradient of the flow matching loss (Goodfellow et al., 2014):
$$\tilde{c} = c + \eta \cdot \text{sign}(\nabla_c \mathcal{L}_{\text{FM}}(c)),$$
where $\eta$ controls the perturbation coefficient. This technique produces pseudo-OOD conditions that remain close to the data manifold but still challenge the model, and thus, improving its ability to distinguish subtle violations of the condition distribution.

### 3.2 OUT-OF-DISTRIBUTION DETECTION

Now that we have established how to train our model (Alg. 1), we need a way to detect OOD conditioning inputs. In the classic flow matching, the interpolated trajectories $x_t$ follow a straight

---

**Algorithm 1** Training *Diverging Flows*

---

**Input:** Training dataset: $D$
**Initialize:** Neural Network $u_t^\theta$
**for** each iteration **do**
    Sample a batch and extract $x_1$ and $c$
    $x_0 \sim p$
    $t \sim \mathcal{U}[0,1]$                                          $\triangleright$ Sample a time step $t$
    $x_t = (1-t)x_0 + tx_1$                     $\triangleright$ Interpolate the two points at time step $t$
    $u_t(x_t) = x_1 - x_0$                              $\triangleright$ Acquire the desired velocity
    $\hat{u}_{t,c} = u_t^\theta(x_t, c)$                                  $\triangleright$ Predict velocity
    $\mathcal{L}_{FM} = \mathbb{E}\|\hat{u}_{t,c} - u_t(x_t)\|^2$                      $\triangleright$ Compute FM loss
    $\tilde{c} = c + \eta \, \text{sign}(\nabla_c \mathcal{L}_{FM})$                   $\triangleright$ Create the OOD condition
    $\hat{u}_{t,\tilde{c}} = u_t^\theta(x_t, \tilde{c})$                     $\triangleright$ Predict velocity with the OOD condition
    $L_{\text{repel}} = \mathbb{E}\big[\max\{\|u_t - \hat{u}_{t,c}\|^2 - \|u_t - \hat{u}_{t,\tilde{c}}\|^2 + \alpha, 0\}\big]$
    $L_{\text{curve}} = \mathbb{E}\big[\max\{(1 - \text{sim}(u_t, \hat{u}_{t,c})) - (1 - \text{sim}(u_t, \hat{u}_{t,\tilde{c}})) + \beta, 0\}\big]$     $\triangleright$ Contrast velocities
    $\mathcal{L}_{\text{DF}} = \mathcal{L}_{\text{FM}} + \lambda \, \mathcal{L}_{\text{repel}} + \gamma \, \mathcal{L}_{\text{curve}}$
    Update $u_t^\theta$ using gradient descent
**end for**

---

path from $x_0$ to $x_1$ with constant velocity $u_t = x_1 - x_0$. Thus, regardless of the quality of the final output, at each step the model should strive to follow this straight trajectory with stable velocity. When the condition is OOD, however, the Diverging Flows model produces velocity fields that either fluctuate around the straight path or diverge completely from it. Quantifying this variability provides a natural score for detecting OOD conditions.

During conditional generation, we collect the predicted velocities into $\hat{U} = \{\hat{u}_{t,\cdot,d}\}_{t=1,\dots,T}^{d=1,\dots,D}$, where $T$ is the number of generation steps and $D$ the number of spatial dimensions. For each dimension $d$, we compute the variance of predicted velocities over time, $\text{Var}_t[\hat{u}_{\cdot,\cdot,d}]$. We then average these variances across the spatial dimensions to obtain the *Flow Smoothness Score (FSS)*:

$$S_{\text{FSS}}(\hat{U}) = \frac{1}{D} \sum_{d=1}^{D} \text{Var}_t[\hat{u}_{\cdot,d}]. \tag{8}$$

A higher FSS indicates greater disturbance in the predicted velocities, which is an indication that the conditioning input is OOD.

To make Diverging Flows practically useful for deployment, we must determine a decision threshold for the FSS. We employ conformal prediction (Gammerman et al., 1998; Angelopoulos et al., 2024) to compute a statistically valid threshold $\tau$ that guarantees a desired coverage level on ID conditions. Specifically, given a calibration set of ID conditions, we query our model, and store the FSS values, $\{S_{\text{FSS}}(\hat{U}_{\text{cal}})\}_{i=1}^{M}$. Then, we set $\tau = \text{Quantile}_{1-\epsilon}\big(\{S_{\text{FSS}}(\hat{U}_{\text{cal}})\}_{i=1}^{M}\big)$. At inference, a condition is declared in-distribution if $S_{\text{FSS}}(\{\hat{u}_{t,c}\}_{t=1}^{T}) \leq \tau$, and OOD otherwise.

### 3.3 IMPLEMENTATION DETAILS

**Dual Conditioning** The way conditioning inputs are introduced to the model plays a significant role in creating diverging flows. The Diverging Flows approach is using a modified version of the U-Net used in the Improved DDPM work (Nichol & Dhariwal, 2021). First, we employ the conditioning used in HyperDM model (Chan et al., 2024). The conditioning input is appended channel-wise to the primary input and reintroduced at the output stage, ensuring that structured information is explicitly available to the decoder. This conditioning is important for the predictive ability of the model. However, the information vanishes during the deep encoder–decoder transformations, and thus, it is not sufficient to align the network's dynamics with the given condition. Hence, we also introduce the condition into the time embeddings. The condition input is processed through a convolutional encoder, projected into the time embeddings' space and then applied using FiLM (Perez et al., 2018). For visualization of the architecture refer to Appendix A.1

**Margin Selection** Two key hyperparameters in our contrastive loss are the margins $\alpha$ and $\beta$. Large margins are essential, since small values lead to saturation and weak gradients, while stronger margins enforce separation and maintain divergence between in- and out-of-distribution flows. In all experiments, we set $\alpha = 100$ and $\beta = 0.8$.

## 4 RELATED WORK

**OOD detection in generative models.** While a large literature exists on out-of-distribution (OOD) detection for discriminative models (Liang et al., 2017; Sun et al., 2022; Li et al., 2023; Du et al., 2022; Wang et al., 2022), there is comparatively less work addressing OOD detection for conditional flow- or diffusion-based models. Prior efforts in unsupervised OOD detection mainly rely on reconstruction-based scores or likelihood-based criteria. However in generative models likelihood-based approaches often fail because deep generative models can assign high likelihood to OOD inputs and low likelihood to in-distribution data (Choi et al., 2018; Nalisnick et al., 2018; Hendrycks et al., 2018; Serrà et al., 2019). Reconstruction-based methods often rely on autoencoders or, more recently, diffusion models to reconstruct inputs and detect anomalies from reconstruction errors. While some works use masking or inpainting within diffusion models, such strategies operate on outputs rather than assessing the validity of the conditioning input itself.

**Diffusion- and flow-based approaches.** More recent methods have explored the dynamics of diffusion models for OOD detection, using statistics computed along the reverse diffusion path (Heng et al., 2024) combined with a statistical model for scoring or reconstruction errors from inpainting with diffusion models (Liu et al., 2023). While these methods are designed for unconditional generation, they do not extend naturally to conditional predictors. Nonetheless, Chan et al. (2024) proposed a hyper-diffusion model for efficient ensemble creation, which was utilized for both prediction and uncertainty quantification. Similarly, several works have applied flow matching models for prediction tasks with some incorporating uncertainty estimation (Zhang et al., 2024; Kollovieh et al., 2024), but none of them address OOD detection.

**Our contribution.** To our knowledge, no prior work extends flow matching with an integrated OOD detection mechanism. Diverging Flows fills this gap by introducing a contrastive objective over instantaneous velocities, paired with a conformal prediction rule, to enable reliable OOD detection without degrading the predictive performance of the underlying flow matching model. Unlike many existing OOD methods that depend on large pretrained diffusion models, our approach is trained simultaneously both for generation and OOD detection, making it applicable even in settings where no pretrained models are available.

## 5 EXPERIMENTS

For the evaluation of Diverging Flows (DF), we begin with a two-dimensional toy example, which provides an intuitive visualization of how contrastive flow matching facilitates OOD detection. We then turn to standard benchmark datasets, which allow for a quantitative comparison with existing OOD detection approaches. Although the goal of Diverging Flows is to unify OOD detection and conditional generation within a single framework, we first isolate the OOD detection component by evaluating the model on a reconstruction task. This setting allows direct comparison with state-of-the-art OOD detection methods that are specifically designed solely for OOD detection. Finally, we demonstrate the applicability of Diverging Flows in a real-world weather forecasting task that requires both accurate prediction and reliable OOD detection via conformal calibration.

We evaluate OOD detection performance using the area under the receiver operating characteristic curve (AUROC) (Fawcett, 2006). Moreover, to assess the quality of reconstructions and predictions, we additionally report mean squared error (MSE) and the structural similarity index measure (SSIM) (Wang et al., 2003; 2004). The hyperparameters used in the experiments are reported at Appendix A.2.

**Datasets** We evaluate Diverging Flows on both RGB and grayscale image datasets. For the RGB setting, we use CIFAR-10 (C10) (Krizhevsky et al., 2009) and SVHN (Netzer et al., 2011) as in-distribution datasets, along with Textures and CIFAR-100 (C100) (Krizhevsky et al., 2009) as OOD

datasets. For the grayscale setting, we consider MNIST (LeCun & Cortes, 2005), FashionMNIST (FMNIST) (Xiao et al., 2017), and KMNIST. In addition, we evaluate Diverging Flows on a real-world weather forecasting task, following the setup of HyperDM (Chan et al., 2024). The task involves predicting surface air temperature maps from the ERA5 reanalysis dataset (Hersbach et al., 2020), consisting of 1,240 samples recorded at six-hour intervals.

**Baselines** We compare Diverging Flows mainly with recent diffusion-based OOD detection approaches, which represent the current state of the art. The generative baselines include LMD (Liu et al., 2023), the two DiffPath variants (Heng et al., 2024), and DDPM-OOD (Graham et al., 2023). For the DiffPath RGB experiments, we follow the authors and use their pretrained model trained on CelebA as the base distribution. For the grayscale experiments, we train DiffPath from scratch on the respective in-distribution dataset. To isolate the contribution of our contrastive objective, we also report ablations of Diverging Flows without the contrastive term, evaluating both likelihood-based and FSS-based (FM-FSS) AUROC. As a non-generative baseline, we include a reconstruction-based Autoencoder (AE). For all baselines, we use the official code released by the authors, except for LMD where we report results directly from the original paper and from the DiffPath work.

## 5.1 TOY EXAMPLE

For our toy example, we design a two-dimensional spiral distribution and sample 10K points, which can be viewed as coordinates on a plane in the range $[-1, 1]^2$. The task for Diverging Flows is the following: given in-distribution coordinates as conditions, the model should transport noise samples to the specified locations while following the optimal straight trajectory as closely as possible. After training, we construct a dense grid of points over the plane and identify those that do not overlap with the spiral distribution (Appendix Fig. 6). These grid points serve as OOD conditions. The objective of Diverging Flows at inference is to detect conditioning inputs that lie outside the training distribution.

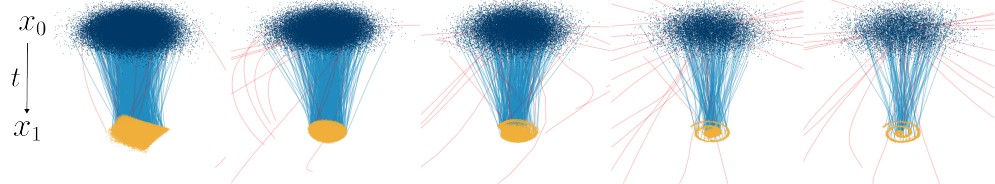

Figure 3: Demonstration of the diverging flow training on a toy example. The blue lines represent the ID conditional generations, whereas the red lines represent the OOD ones, as detected by the conformal threshold. Left to right: training progress from epoch 20 to 5000, showing increasingly sharper separation.

In the 2D spiral toy task, Diverging Flows quickly learns to separate valid from invalid conditions. The ID inputs produce smooth, near-linear transport trajectories, while OOD inputs yield irregular, fluctuating flows that are easily detected via the FSS. Using the conformal calibration, we achieve an AUROC of 0.991.

## 5.2 OUT-OF-DISTRIBUTION DETECTION RESULTS

The OOD detection results across benchmarks are reported in Table 1. For training Diverging Flows, we use 10K samples from each dataset due to computational constraints. In the case of the Textures dataset, images are resized to match the resolution of the in-distribution datasets.

The results show that Diverging Flows outperforms the state-of-the-art diffusion-based OOD detection methods, achieving perfect AUROC scores on the majority of benchmarks. Additionally, we should highlight that it is the only method to reach such high performance on the challenging CIFAR-10 vs. CIFAR-100 task, which is practically a near-OOD task. For the CIFAR-10 task, we illustrate the FSS histogram at Fig. 2.

Table 1: Comparison of various baselines across different ID and OOD datasets. We report the AUROC of each approach. Higher is better. **Bold** represents the best result across the baselines and second best is underlined. In instances, where multiple baselines are tied, all of them are marked.

| Methods | C10 vs | | SVHN vs | | MNIST vs | | KMNIST vs | | FMNIST vs | |
| --- | --- | --- | --- | --- | --- | --- | --- | --- | --- | --- |
| | SVHN | Textures | C100 | C10 | Textures | FMNIST | KMNIST | MNIST | FMNIST | MNIST | KMNIST |
| AE-MSE | 0.305 | 0.520 | 0.581 | 0.934 | 0.846 | 0.987 | 0.975 | 0.508 | 0.608 | 0.971 | 0.990 |
| FM Likelihood | 0.400 | 0.500 | 0.500 | 0.700 | 0.652 | 0.214 | 0.133 | 0.614 | 0.138 | 0.652 | 0.609 |
| DiffPath-1D | 0.956 | 0.685 | 0.551 | 0.971 | 0.949 | 0.988 | 0.863 | 0.699 | 0.420 | 0.818 | 0.696 |
| DiffPath-6D | 0.911 | 0.922 | 0.590 | 0.939 | 0.981 | **1.00** | 0.949 | 0.885 | **1.00** | **1.00** | **1.00** |
| LMD | 0.922 | 0.667 | 0.604 | 0.919 | 0.914 | 0.999 | 0.984 | **0.978** | 0.993 | 0.992 | 0.990 |
| DDPM-OOD | 0.390 | 0.598 | 0.536 | 0.951 | 0.910 | 0.997 | 0.996 | 0.770 | 0.518 | 0.924 | 0.908 |
| **Ours** | | | | | | | | | | | |
| FM-FSS | 0.973 | 0.902 | 0.264 | 0.005 | 0.196 | 0.996 | 0.972 | 0.406 | 0.992 | 0.788 | 0.806 |
| DF + Noise | 0.924 | 0.999 | 0.313 | **1.00** | **1.00** | **1.00** | **1.00** | 0.301 | 0.905 | 0.453 | **1.00** |
| DF + FGSM | **1.00** | **1.00** | **0.938** | **1.00** | **1.00** | **1.00** | **1.00** | 0.502 | **1.00** | 0.802 | **1.00** |

A performance drop is observed, however, in the KMNIST vs. MNIST setting. We speculate that the problem originates from the strong visual similarity between the two datasets and the fact that MNIST digits typically occupy a smaller spatial area than KMNIST digits. As a result, MNIST conditions resemble "underfilled" KMNIST samples rather than clearly distinct OOD inputs, making them borderline in-distribution.

Considering the creation of pseudo-OOD conditions during training, FGSM perturbations consistently result in stronger performance than Gaussian noise. This is expected, as FGSM generates more challenging near-OOD samples, enabling the model to learn a sharper representation of the condition distribution. Nevertheless, Gaussian noise remains a viable choice for tasks where a simpler separation is sufficient.

The FM-FSS baseline highlights that dual conditioning alone can influence the flow dynamics, especially in tasks where ID and OOD samples are easily distinguishable. However, in more difficult scenarios, FM-FSS is less effective, and competitive results are only achieved with the addition of the contrastive objective.

In Table 2, we also report reconstruction quality for Diverging Flows compared to vanilla flow matching. The results show that incorporating the contrastive objective not only preserves reconstruction quality (Appendix A.4) but also improves it according to SSIM. This improvement is expected due to the effect of the triplet margin loss, which provides an additional training signal by pulling positive samples closer to the anchor. Next, we present the weather forecasting task, where we demonstrate how Diverging Flows can be combined with conformal prediction to deliver both accurate predictions and reliable OOD detection.

Table 2: Comparison of the reconstruction ability between Diverging Flows and flow matching training.

| | C10 | | SVHN | | MNIST | | KMNIST | | FMNIST | |
| --- | --- | --- | --- | --- | --- | --- | --- | --- | --- | --- |
| | MSE $\downarrow$ | SSIM $\uparrow$ | MSE $\downarrow$ | SSIM $\uparrow$ | MSE $\downarrow$ | SSIM $\uparrow$ | MSE $\downarrow$ | SSIM $\uparrow$ | MSE $\downarrow$ | SSIM $\uparrow$ |
| DF | **0.043** | **0.994** | 0.033 | **0.996** | **0.003** | **0.983** | 0.002 | **0.983** | 0.002 | **0.972** |
| FM | 0.045 | 0.988 | **0.030** | 0.994 | 0.011 | 0.925 | **0.001** | 0.944 | **0.001** | 0.941 |

## 5.3 WEATHER FORECASTING

Weather forecasting is regarded as a critical application, where diffusion models have been quite successful (Price et al., 2025; Andrae et al., 2025). In our weather forecasting task, we train both Diverging Flows and HyperDM (Chan et al., 2024) using $64 \times 64$ resolution images. Using the ERA5 dataset (Hersbach et al., 2020), the objective is to predict surface temperature heatmaps six hours into the future, given the current map. We construct OOD cases by following the HyperDM experiments,

which introduces an artificial hotspot in a region that typically exhibits low temperatures in the training set. From the training data, we also reserve a calibration split for computing the conformal prediction bound. The $\epsilon$ parameter can be fine-tuned to provide the best possible results. In our case, we use an $\epsilon = 0.05$.

Table 3: Weather forecasting results using the ERA5 dataset. The models predict the temperature heatmap six hours in the future, given the current one. We report the MSE, SSIM, AUROC, and Conformal AUROC (AUROC after conformal calibration to ensure statistically valid thresholds).

|  | MSE ↓ | SSIM ↑ | AUROC ↑ | Conformal AUROC ↑ |
|---|---|---|---|---|
| HyperDM | 0.014 | 0.954 | — | — |
| DF (**Ours**) | $5 \times 10^{-5}$ | **0.965** | 0.999 | 0.969 |

The results of the weather forecasting indicate that Diverging Flows is highly performant in prediction tasks. Some examples of the predictions are shown in Fig. 4. Regarding the OOD detection, the method attains near-perfect AUROC. A slight decrease in performance is observed when applying conformal prediction, but this reduction is negligible given the benefit of a statistically valid threshold that enables reliable, real-time OOD detection in real-world scenarios.

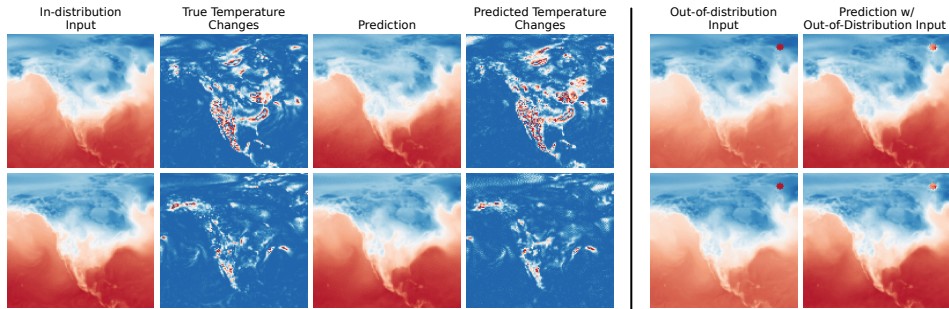

Figure 4: Prediction results on two samples of weather forecasting. We show how both ID and OOD inputs are predicted by the flow model. We also present how the ground truths and predictions differ from the input to better capture the expected result.

## 6 CONCLUSION

In this work, we introduced Diverging Flows, a contrastive flow matching framework that extends vanilla flow matching with built-in out-of-distribution detection in conditional generation. Our approach preserves the generative quality and all core properties of standard flow matching, while additionally providing a reliable signal to identify OOD conditions. As such, Diverging Flows can serve as a drop-in replacement for flow matching in most applications, combining strong generation with robust OOD awareness.

Through experiments on benchmark datasets and a real-world weather forecasting task, we showed that Diverging Flows achieves almost perfect OOD detection performance while maintaining strong generative capabilities. Diverging Flows directly targets the conditioning mechanism of flow models by exploiting the velocity fields as discriminative signals. This intrinsic coupling between generation and OOD detection explains the near-perfect AUROC on challenging benchmarks such as CIFAR-10 vs. CIFAR-100. This highlights the potential of Diverging Flows as a reliable tool for safe conditional prediction in high-stakes domains.

The main limitations are that training with the contrastive objective can be unstable and requires careful tuning of the margins and loss weights, and that generating pseudo-OOD conditions with FGSM introduces additional computational overhead due to the extra gradient step. As for future directions, we will explore how we can use the dynamics of the flow models for uncertainty quantification, how to incorporate these components into pretrained models and apply Diverging Flows in complex high-stakes robotics applications.

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

## A  APPENDIX

### A.1  DUAL CONDITIONING DETAILS

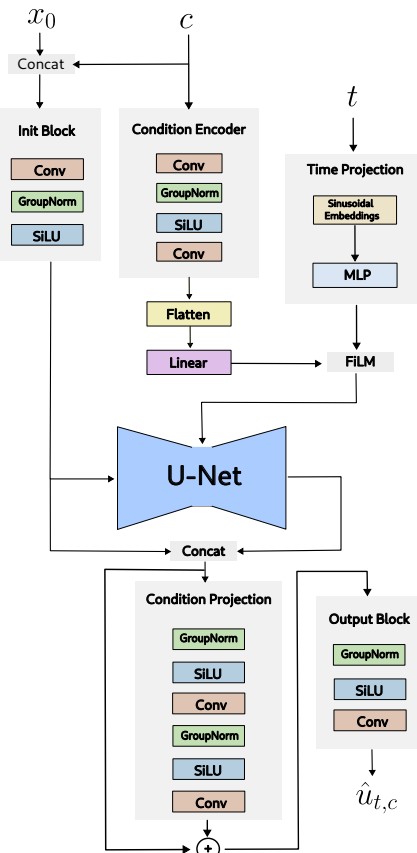

Figure 5: Dual Conditioning architecture visualization.

In Figure 5, we illustrate the dual conditioning mechanism used in our U-Net model. The model receives three inputs: the noise $x_0$, the condition $c$, and the time variable $t$. The noisy input is concatenated with the condition and processed through an initial block that consists of a convolution layer, normalization, and activation function. The normalization that is applied is implemented with Group Normalization, which normalizes features across groups of channels as

$$\text{GroupNorm}(x) = \frac{x - \mu_g}{\sqrt{\sigma_g^2 + \epsilon}} \cdot \gamma + \beta,$$

where $\mu_g, \sigma_g$ are computed per group. The number of groups used in the experiments is 32. The non-linearity is the SiLU (Swish) activation,

$$\mathrm{SiLU}(x) = x \cdot \sigma(x) = \frac{x}{1 + e^{-x}}.$$

Then the condition $c$ is further processed by a dedicated encoder, then flattened and projected to produce scaling and shifting parameters $[s, \delta] \in \mathbb{R}^m$. The time variable $t$ is embedded using sinusoidal features, which are then mapped through an MLP to yield a time embedding $e_t$. Both condition and time embeddings are combined to modulate the U-Net feature maps through FiLM

$$h' = s \odot h + \delta,$$

where $h$ is the intermediate feature map and $\odot$ denotes elementwise multiplication.

At the decoder side, the output of the U-Net is concatenated with the output of the Init Block. The result then passes through a residual block and its output serves as the input to final output block, which produces the predicted velocity field $\hat{u}_{t,c}$. This design provides *dual conditioning*: (i) through the time-condition embedding, which is applied across all U-Net blocks, and (ii) reintroduction of the condition at the input and output. Together, these pathways preserve the generative quality under in-distribution inputs while encouraging divergence under out-of-distribution conditions.

## A.2 HYPERPARAMETERS

In Table 4, we present the hyperparameters used in our experiments.

Table 4: List of Hyperparameters used in our experiments.

| **Hyperparameters** | |
| --- | --- |
| **Model Layers** | |
| Convolution Kernels | $3 \times 3$ |
| GroupNorm Channels | 32 |
| Embedding Size | 256 |
| Number of Downsampling Blocks | 3 |
| Number of Middle Blocks | 2 |
| **Model Training** | |
| Epochs | 10,000 |
| Batch Size | 64 |
| Optimizer | AdamW |
| Learning Rate | $1 \times 10^{-4}$ |
| Weight Decay | $1 \times 10^{-2}$ |
| Learning Rate Scheduler | Linear |
| Scheduler Start Factor | 1 |
| Scheduler End Factor | $1 \times 10^{-4}$ |
| EMA Rate | 0.999 |
| **FGSM** | |
| Perturbation Coefficient $\eta$ | 0.3 |
| **Losses** | |
| Margin $\alpha$ | 100 |
| Margin $\beta$ | 0.8 |
| Weight $\lambda$ | 0.2 |
| Weight $\gamma$ | 0.1 |

## A.3 2D TOY EXAMPLE DATA

In Figure 6, we illustrate the data points used for the evaluation of the toy example.

## A.4 RECONSTRUCTION EXAMPLES

In Figures A.4 and A.4, we present some reconstructions produced by the Diverging Flows model.

Figure 6: Visualization of the 2D points used in the toy task.

CIFAR-10 SAMPLES

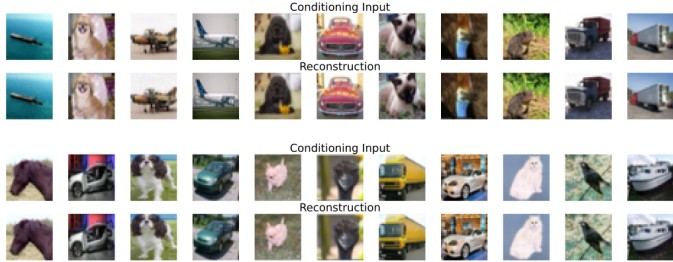

SVHN SAMPLES

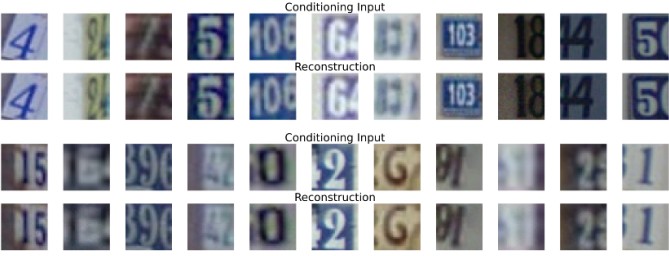

