# OpenReview forum: "Diverging Flows: Detecting Out-of-Distribution Inputs in Conditional Generation"
_ICLR.cc/2026/Conference — Submitted to ICLR 2026_

### Official Review · Reviewer_aXV4 · 2025-10-24

**Soundness:** 2
**Presentation:** 3
**Contribution:** 2
**Rating:** 4
**Confidence:** 3

**Summary:**

This paper proposes Diverging Flows, a flow-matching approach for conditional generation that can detect OOD conditions. The method augments flow matching with a contrastive objective, ensuring that predicted velocity fields for ID and OOD conditions are separated during training. Experiments include a 2D toy task, image reconstruction OOD benchmarks, and a weather-forecasting setting. Results suggest high OOD detection performance while preserving or slightly improving reconstruction quality.

**Strengths:**

S1. The direction of calibrating uncertainty in generative models is important for making conditional generation safer and more reliable.

S2. The technical components are clearly described and easy to follow.

S3. The toy example in Section 5.1 clarifies the core concept and how divergence manifests along trajectories.

S4. The paper candidly reports limitations of the proposed approach.

**Weaknesses:**

W1. The problem setting feels somewhat removed from common practical goals. In many applications, diffusion or flow models are expected to generalize to novel conditions like "an astronaut riding a horse on mars"; this work instead trains a generator that intentionally fails under OOD conditions to flag them. The paper would benefit from concrete scenarios where this behavior is clearly advantageous.

W2. Prior work, DiffPath, has already used diffusion or flow trajectories for OOD detection. This paper can be read as an extension of that idea to flow matching with conditional generation; the incremental novelty feels modest.

W3. Some comparisons appear potentially unfair. In Table 1, the DiffPath baseline uses a single model trained on CelebA for RGB experiments, while the proposed method is trained separately on each dataset.

**Questions:**

Q1. Why does the triplet-style component in your loss also improve reconstruction quality? An intuitive explanation would help.

Q2. Can you offer any theoretical guarantee or clear conditions under which the contrastive training should improve OOD detection without hurting generation?

Q3. The introduction motivates safety-critical settings such as weather forecasting and robotics. Under OOD inputs, however, the model is designed to produce diverging flows; what prevents these predictions from being harmful in practice, and how is risk actually mitigated at deployment time?

Q4. For Table 2, are reconstruction errors computed on a held-out test set that was not seen during training?

Q5. Compared with DiffPath, what are the pros and cons in terms of the practical usage (e.g. runtime)?

Q6. In Table 1, would it be possible to compare against DiffPath models trained individually on each dataset to ensure a fairer baseline?

---

> ### Author Response · Authors · 2025-11-28
>
> We would like to thank the reviewer. We expand on their concerns below:
>
> - W1: We agree that in creative applications, "generalization" to novel concepts is desirable. However, our work targets **safety-critical predictive tasks** (e.g., weather, medical), where fidelity to ground truth is required. In these domains, "generalizing" to OOD inputs (e.g., sensor failure) leads to **hallucination,** plausible but factually incorrect predictions. Standard flow models lack a mechanism to quantify this hallucination. Diverging Flows fills this gap not by preventing the model from functioning, but by equipping it with a "telltale sign" (the OOD score). This allows downstream systems to **abstain** from making predictions on unreliable inputs, rather than forecasting plausible but erroneous events.
> - W2:  While both methods leverage trajectory dynamics, **Diverging Flows** fundamentally differs from and outperforms DiffPath (see Table 1):
>     1. DiffPath analyzes trajectories of frozen models, **only in the diffusion setting**. We actively optimize the velocity field via a contrastive objective to *enforce* divergence for OOD inputs, rather than merely observing naturally occurring deviations.
>     2. DiffPath targets unconditional generation. We address the harder problem of validating high-dimensional conditions, implicitly learning the valid manifold by contrasting against perturbations.
>     3. DiffPath is a dedicated detector. Our method performs generation and detection simultaneously in a single forward pass, providing the OOD signal as a computational byproduct without requiring separate models.
> - W3 and AQ6: We agree that fair baselines are essential. For RGB, we used the official DiffPath pipeline for reproducibility. However, to explicitly address the concern about training disparities (Q6), our **Grayscale experiments** (Table 1) were designed as a controlled comparison where we trained DiffPath **from scratch** on the exact same datasets. Even in this identical setting, Diverging Flows consistently outperforms DiffPath (e.g., 0.905 vs. 0.420 AUROC on FMNIST vs. KMNIST). This confirms that our advantage originates from the contrastive objective, not pre-training differences.
> - AQ1: The improvement stems from the regularization effect of the contrastive objective. By explicitly penalizing velocity fields that originate from OOD inputs (Eq. 5 & 6), we force the model to learn a sharper, more robust representation of the optimal velocity for ID data. This mirrors margin-based metric learning: "pulling positive samples closer" (aligning valid flows) is a natural consequence of pushing negatives away, resulting in better reconstruction fidelity.
> - AQ2: While we do not provide a formal theorem, the optimization objective itself (Eq. 5 & 6) acts as a mathematical condition for divergence. Assuming the training converges, the loss enforces that $||u_{t, ID} - u_{t, OOD}||^2 \geq \alpha$. Consequently, as long as the pseudo-OOD samples sufficiently cover the "boundary" of the valid condition manifold, the model is forced to distinguish them. The generation quality remains preserved because the primary term $\mathcal{L}_{FM}$ (Eq. 2) ensures the vector field for ID inputs still interpolates $p$ to $q$ correctly, while the contrastive terms only activate when flows fail to separate
> - AQ3: The "diverging flow" is an internal signal, not the final output. In deployment, the FSS acts as a flag: we compare it against a calibrated threshold $\tau$ (derived via Conformal Prediction). If $FSS > \tau$, the system flags the input as unreliable and does not proceed to prediction. Thus, risk is mitigated by detecting the divergence and halting downstream action before a potential hallucination can be utilized.
> - AQ4: Yes, for all datasets in Table 2 (and Table 1), reconstruction errors and OOD metrics are computed on the standard test/validation splits (e.g., the official CIFAR-10 test set, ERA5 test split), which were never seen during training.
> - AQ5: Pros and cons vs. DiffPath
>     - **Pros:** Diverging Flows is significantly more efficient. Since DiffPath is designed to analyze the trajectory of the generated variable, applying it to validate an *input condition* effectively requires a costly two-step pipeline: first running a diffusion process on the input to check its validity, and only *then* running the separate predictor. In contrast, our method unifies these tasks: the validity of the condition is assessed *simultaneously* during the prediction of the target, yielding the OOD signal as a "free" byproduct of the single forward pass.
>     - **Cons:** The main trade-off is that Diverging Flows requires training the model from scratch with the contrastive objective to embed this capability, whereas DiffPath can often be applied to pre-trained models (with the operational limitations noted above). We acknowledge this in our Limitations section.

---

### Official Review · Reviewer_jvGi · 2025-10-26

**Soundness:** 2
**Presentation:** 3
**Contribution:** 2
**Rating:** 2
**Confidence:** 4

**Summary:**

This work proposed a new method based on (conditional) flow matching to detect out of distribution (OOD) data, using only in-distribution (ID) training data. The main idea is to use the variance of the vector field at multiple time steps (learned through a contrastive variant of flow matching) as a scoring function. Experiments on standard image benchmarks and a weather forecasting dataset seem to confirm the effectiveness of the proposed method.

**Strengths:**

- a new OOD detection algorithm based on (conditional) flow matching and contrastive learning

- experiment comparison against multiple diffusion based OOD detection baselines on standard image benchmarks

**Weaknesses:**

- unclear how the context for in-distribution data is generated, on both training and test sets.

- problem setup is questionable: why insisting on training only on in-distribution data? Anyone who is serious about OOD should at least try to collect a small amount of OOD data, for otherwise we are doing ourselves a disservice and making the problem unnecessarily underdefined and challenging. I recognize many prior works followed the same setup and I am curious to hear the authors' reasons. For some theoretical discussions on this problem setup, see e.g. https://openreview.net/forum?id=sde_7ZzGXOE and https://proceedings.mlr.press/v139/zhang21g.html.

- heavier computational cost for both training and inference: for training, one needs to use a bigger network to account for the context, while for testing, multiple time steps need to be simulated and averaged. Can the authors comment on scalability and training and test time comparisons? Ideally, one should compare to the unconditional flow matching model on the amount of in-distribution data needed to achieve certain performance (e.g., the assumption on Line 052).

**Questions:**

My main concern is that the authors did not describe how the context c during training and testing is generated and how its choice (including dimension) affects the experimental results. This is a crucial missing piece that will affect my final evaluation.

I also suggest running this ablation study: during testing, what happens if we disable the context (e.g., provide the same context for all test samples)? Can your method still achieve good performance? In other words, is the context actually doing the heavy lifting?

It is surprising that in Table 1, DF+FGSM achieved AUROC 1 on multiple datasets, implying that those pairs of datasets have no overlap of support at all. If this is the case, many existing generative methods (including those based on likelihood) should perform well too. To the contrary, FM likelihood performed very poorly. Do the authors have any explanation of this observation? How is FM likelihood trained and tested?

In the second paragraph of Introduction, the authors made a big deal on quantifying uncertainty, but in the end the authors merely employed (split) conformal prediction to address this issue. This is a bit disappointing since the same (split) conformal prediction could literally be applied to any existing OOD detection method.

---

> ### Author Response · Authors · 2025-11-28
>
> We thank the reviewer for the comments and address each point below.
>
> - W1 and  AQ1: We apologize if the definition of the context was unclear. To clarify: in our conditional generation tasks (e.g., reconstruction, forecasting), the context $c$ is simply the **observed input data** (e.g., the current weather map or the image to be reconstructed).
>     - **Training:** Pairs of $(x, c)$ are drawn directly from the training dataset as mentioned in the Algorithm 1 (e.g., $x$ is the target temperature map, $c$ is the current map).
>     - **Inference:** The model receives a test condition $c_{test}$ and generates the prediction.
>     - **Effect of Choice/Dimension:** The dimension of $c$ is determined by the task (e.g., $64\times64$ images).
> - W2:  Training exclusively on ID data is intentional and follows the standard OOD-detection setting established in the literature [1], where real OOD samples are not assumed to be available. In practice, we do not know what the true OOD distribution looks like. The defining property of OOD data is precisely that it lies outside the support of the training distribution. Since its structure is unknown, collecting "representative" OOD samples is not well-defined and can bias the model to detect only "known unknowns" while failing on novel anomalies [2]. For the same reason, even the notion of "in-distribution" is only approximated by the dataset; we never have access to the full underlying data manifold, only the finite subset provided by the training set. Because of this fundamental asymmetry, our methodology uses pseudo-OOD perturbations to perform controlled off-manifold exploration. Our pseudo-OOD examples do not aim to model real OOD distributions, but simply encourage the model to observe how the conditional velocity field behaves when the conditioning variable moves away from the data manifold. This yields a reliable divergence signal without making assumptions about the structure of true OOD inputs.
> - W3: There is a slight misunderstanding regarding inference cost. Flow Matching always requires solving an ODE (typically 10-50 steps). Our Flow Smoothness Score (FSS) is computed **simultaneously** during this loop by measuring the variance of the velocities already being generated. We do not run additional simulations; thus, inference time is effectively identical to standard FM. Similarly, the training overhead (dual encoder + contrastive loss) is negligible compared to the U-Net backpropagation.
> - AQ2: At inference, the test sample itself is the conditioning variable. The model receives noise along with this condition and transports the noise according to the velocity field induced by the conditioning. If the conditioning is replaced by noise, zeros, or any non-informative pattern, it moves off the data manifold, and the resulting velocity field no longer matches the behavior learned for valid ID conditions. This produces the same divergence effect as our pseudo-OOD training conditions and yields a high OOD score. In short, the conditioning determines whether the model stays on-manifold, so corrupting it naturally leads to OOD detection.
> - AQ3: FGSM produces "near-manifold" perturbations designed to strongly disrupt the velocity field. Because FSS measures deviation from the stable ID field, these targeted perturbations generate a consistent divergence signal, yielding high separability. When structure overlaps (e.g., MNIST/KMNIST), AUROC naturally decreases.
> - Regarding the FM Likelihood:  We confirm that FM Likelihood refers to the standard flow-matching likelihood estimated via the ODE trace (Hutchinson's estimator)[3]. As noted in our paper and widely cited in literature[4,5], deep generative models (Flows, VAEs) frequently assign **higher likelihoods** to simpler OOD datasets (e.g., SVHN) than to complex ID datasets (e.g., CIFAR-10) due to the dominance of low-level statistics (background, constant colors) over semantic content. Our results confirm this well-known phenomenon, justifying the need for velocity-based metrics like FSS.
> - AQ4: We agree that Conformal Prediction (CP) is a wrapper applicable to any method. We do not claim CP as a novelty; rather, we use it to demonstrate that our Flow Smoothness Score is well-behaved enough to be calibrated. Many OOD scores are unstable or multimodal, making CP difficult. Our results show that Diverging Flows yields a score distribution that enables statistically valid decision boundaries (Table 3), which is the ultimate requirement for the safety-critical applications we target.
>
> [1] Hendrycks, D., & Gimpel, K. A baseline for detecting misclassified and out-of-distribution examples in neural networks.
>
> [2] Hein et al. Why ReLU networks yield high-confidence predictions far away from the training data.
>
> [3] Lipman et al., “Flow Matching Guide and Code”.
>
> [4] Nalisnick et al., “Do Deep Generative Models Know What They Don’t Know?”
>
> [5] Kirichenko et al., “Why Normalizing Flows Fail to Detect Out-of-Distribution Input.”

---

### Official Review · Reviewer_CdUW · 2025-10-31

**Soundness:** 1
**Presentation:** 3
**Contribution:** 2
**Rating:** 2
**Confidence:** 4

**Summary:**

The authors propose a framework which simultaneously generates conditional samples and also determine if the conditioning is out of distribution. They do this by creating Gaussian noise or adversarial perturbations of real samples and use a contrastive loss to push OOD conditioned samples away from in-distribution samples. At test time, the instability in the curve of the predicted velocity field grants a Flow Smoothness Score metric which, once surpassing a certain threshold which is obtained through conformal prediction, allows the model to flag whether the sample is in-distribution or OOD. They evaluate on a few standard image datasets which contrast against one another.

**Strengths:**

Strengths:
Their approach allows them to train a single model for OOD detection without specific OOD data needed within the training procedure.

They show surprisingly strong results in terms of AUROC (see weaknesses section for a caveat here).

Generation quality seems to not be impacted by the inclusion of the OOD detection algorithm.

**Weaknesses:**

Weaknesses:
The AUROC numbers reported are extremely close to 1 on all the benchmarks. This might be due to the datasets being completely different when trying to direct OOD samples, whereas in the real world the application may not be as clean. In particular, the MNIST vs KMNIST case shows a severe drop in performance which is irregular for a model which should otherwise be performing very well across everything else. This might indicate that the model is not actually robust to OOD detection as claimed.

Some ablation study results regarding the inclusion of the dual-conditioning portion vs the contrastive loss could be helpful in determining the specific contributions of the paper.

Datasets included are somewhat weak; the weather forecasting is done for single step only. Current weather forecasting experiments in particular focus on much longer time horizons and multi-step prediction.

The authors mention in the limitations that the training with the contrastive loss can be unstable, but fail to mention how heavy the impact of the instability is, which can make applying this model particularly hard.

**Questions:**

See weaknesses.

---

> ### Author Response · Authors · 2025-11-28
>
> We would like to thank the reviewer for their comments.
>
> - W1: We respectfully disagree with the interpretation that the drop in performance on the KMNIST vs. MNIST task indicates a lack of robustness; rather, it confirms the soundness of our method. The near-perfect scores on distinct datasets (e.g., SVHN vs. CIFAR) are expected because our contrastive objective explicitly trains the velocity field to diverge when inputs deviate from the manifold. For semantically distinct datasets, this divergence naturally maximizes. However, KMNIST shares significant structural similarities with MNIST; MNIST digits essentially look like "incomplete" or simpler versions of KMNIST characters. Consequently, this is a **near-distribution** task rather than a clear OOD task. Table 1 confirms that this difficulty is intrinsic to the dataset pair rather than our method: all baselines exhibit a severe performance drop on this specific setting compared to other benchmarks (e.g., DiffPath drops to 0.699, and AE-MSE drops to 0.508, performing similarly to our method). The fact that detection performance drops for all methods indicates that our model is correctly capturing the semantic ambiguity of this overlap, rather than overfitting to superficial artifacts that would allow artificial separation.
> - W2: The reviewer requests an ablation to determine the specific contributions of the dual-conditioning vs. the contrastive loss. **This ablation is already provided in Table 1 under the name "FM-FSS"**.  This baseline represents the model with **Dual-Conditioning only** (standard Flow Matching training), using the Flow Smoothness Score for detection. By comparing **FM-FSS** (Dual-Conditioning only) with **DF+FGSM** (Dual-Conditioning + Contrastive Loss), we isolate the exact gain from our proposal. For example, on C10 vs. C100, FM-FSS achieves only 0.264 AUROC, while adding the contrastive loss boosts this to 0.938. This effectively demonstrates that dual-conditioning alone is insufficient for hard OOD tasks, and the contrastive loss is the primary driver of performance.
> - W3: Our choice of single-step forecasting is deliberate and follows the standard setup for evaluating conditional generative models in this domain (e.g., HyperDM). Our objective is to evaluate the **OOD sensitivity of the velocity field** at the point of prediction. Multi-step forecasting introduces autoregressive error accumulation, which conflates "OOD detection" with "error propagation stability." Validating the method on single-step prediction is the foundational requirement. In a real-world multi-step rollout, our OOD detector would simply be applied at each step $t$ to ensure the input from $t-1$ is still valid before proceeding.
> - W4: We appreciate the opportunity to clarify the "instability" mentioned in our limitations. We do not mean that training diverges or gradients explode. Rather, we refer to the sensitivity of margin hyperparameters ($\alpha, \beta$) typical of triplet-based losses. If margins are set too low, the loss saturates; if too high, it dominates the generation loss. However, as noted in Appendix A.25, we found a stable configuration ($\alpha=100, \beta=0.8$) that works consistently across datasets.

---

### Official Review · Reviewer_DmnZ · 2025-10-31

**Soundness:** 2
**Presentation:** 3
**Contribution:** 2
**Rating:** 4
**Confidence:** 3

**Summary:**

This paper aims to detect out-of-distribtuons (OOD) conditions in a conditional generative model (here, in the flow matching framework).

The proposed method, called Diverging Flows, seeks to identify OOD conditionings without relying on the model’s output, but instead by computing scores directly from the velocity fields of the flow matching model.
It relies on *contrastive learning*, i.e. it adapts the standard flow matching loss by adding 2 regularization terms.
These two terms aim at pulling apart positive conditionings from negative ones.

The paper suggests two strategies to build negative samples: either pure noise or adversarial samples. Thus, it does not rely on an extra OOD dataset.
The training objective encourages the model to produce similar instantaneous velocity fields for in-distribution (ID) conditionings and divergent velocity fields for OOD conditionings.

The OOD detection then relies on an ad-hoc score that measures the variation of the velocity field along the ODE trajectory: the more it varies, the less likely the conditioning is ID.

**Strengths:**

- The OOD detection problem is interesting.
- The presentation is clear, making the paper easy to follow.
- The method shows empirical effectiveness, and it is applied to real data (weather forecasting).

**Weaknesses:**

**On the OOD Score**
In Section 3.2, I think there is a confusion between:
- what happens during training where one regresses against the conditional velocity field $u^\mathrm{cond} = x_1 -x_0$ on interpolated $x_t$
- what happens during inference, where the goal is to follow the total velocity field which is defined as $u_t = \mathbb E[u^\mathrm{cond}  | x_t]$.

The sentence ''at each step the model should strive to follow this straight trajectory'' is confusing: at inference, the trajectories have no reason to be straight (see e.g. Multisample Flow Matching: Straightening Flows with Minibatch Couplings, Pooladian et al. or Flow Straight and Fast: Learning to Generate and Transfer Data with Rectified Flow, Lie et al. for a discussion on straightness).
As the OOD score is based on a straightness hypothesis; this should be clarified in the paper.

**On conformal prediction** The authors use the term conformal prediction at several places in the paper to explain that their method can be used on ''safety-critical'' usages and that it is ''statistically valid''.
From my understanding, the threshold on the scores (that decides if the data is ID or OOD) is set as a quantile on a calibration set made of ID data.
I think this does not bring any guarantee on what happens for OOD data.
Besides, as it is presented as an important asset of the method, I think this part deserves more details both regarding the theory (do we have some guarantess? if yes, they should be described) and in practice (e.g. in the experiments, what is the size of the calibration set?).

**On generation performance** The claim that the modified loss “does not degrade the predictive performance of the underlying FM model” could be more deeply evaluated. For example, Table 2 reports a reconstruction error, but as this is a generative model, computing the FID is also an important metric to assess the quality of generated samples.

**Questions:**

1. In the experiment made Section 5.2, ID conditionings are exactly samples from the training data.
As stated in the paper, the method is somehow underemployed in this setting (it's rather used as a first benchmark against other baselines). Yet, it would be interesting to add tasks such as image restoration or style transfer (where the degraded images / the styles are the conditionings), it would be a more relevant demonstration of the method's utility.

2. Does the proposed loss really fit into the standard contrastive learning framework ?
Usually, positive samples are generated by applying well-chosen transformations to true samples, while negative samples correspond to other samples from the train set. Here, there is no positive samples (just the baseline sample) and the negative samples are new conditionings. I think this distinction deserves some clarification.

3. For the weather forecasting experiments, are the same hyperparameters and architecture used as in the RGB image experiments? What are the sizes of the training, calibration, and test sets? This information should be provided in the appendix.

**Minor comments**

- The introduction to Flow Matching (FM) could be slightly clarified: $u_t^\mathrm{cond}= x_1-x_0$ is the conditional velocity field  (conditonned on $(x_0,x_1)$) and what we actually aim to learn is $u_t = \mathbb E[u^\mathrm{cond}  | x_t]$.

- On the beginning of section 3, introducing conditional generation. The fact that it is enough to train a velocity field $u_\theta(x, c)$ to generate $q(x|c)$ using the standard FM loss comes from the hypothesis made in Equation (4): the underlying assumption is that $p_t(x_t|x_1)$ is independant of $c$ conditionnaly on $x_1$, which leads to Equation (3). As it is now stated (with Eq (3) that appears before Eq (4)), it is not so clear.

---

> ### Author Response · Authors · 2025-11-28
>
> First, we would like to thank the reviewer for the constructive feedback
>
> - W1: Regarding the comment on the OOD score, we agree that the phrasing in Section 3.2 was unclear. Our method does not assume that inference trajectories must be straight. Indeed, at inference, the model follows its learned conditional field, which may or may not be linear. Thus, the OOD score is not based on enforcing straightness of the generated path. What matters is the stability of the instantaneous velocity field. Under in-distribution conditions, the model is trained to keep $u_t^\theta(x_t, c)$ close to the supervised target velocity. Under the pseudo-OOD conditions, the contrastive losses explicitly push  $u_t^\theta(x_t,\tilde{c})$ away from the in-distribution field. The resulting fluctuation over time becomes a reliable discriminative signal. In other words, the triplet-margin losses are not trying to force OOD trajectories to diverge from the “straight OT path,” but to diverge from the in-distribution conditional velocity fields. We will update our phrasing accordingly in the revised version.
> - W2: To clarify the use of conformal prediction: in standard conformal regression, given calibration pairs $(x_i, y_i)$, the conformity scores are defined as $s_i = |y_i - \hat{f}(x_i)|$, and for a target coverage level $1-\varepsilon$ the conformal threshold is $\tau = \mathrm{Quantile}_{1-\varepsilon}(\{s_i\})$.  The resulting prediction interval for a new input $x$ is $[\hat{f}(x) - \tau,\, \hat{f}(x) + \tau]$, with the usual guarantee $\Pr(|Y - \hat{f}(X)| \le \tau) \ge 1-\varepsilon$  for $(X,Y)$ drawn from in-distribution.  In our setting, the quantity we score is a non-negative statistic $S(c)$,  whose ideal in-distribution value is zero, which makes the implicit ground truth equal to zero. As a result, the conformal prediction reduces to quantile calibration of this statistic, producing a single upper threshold $S(c) ≤ τ$, which is what is required for OOD detection. The standard coverage guarantee continues to apply for in-distribution conditions, while no guarantee is expected for OOD inputs. We will explicitly clarify this reduction to quantile calibration in the revised manuscript.
> - W3: We thank the reviewer for raising this point regarding generation metrics. We respectfully clarify that while FID is standard for synthesis (where diversity is important), it is unsuitable for the conditional prediction tasks we target. In these paired settings, the goal is fidelity to a specific ground truth. FID implicitly rewards diversity; in safety-critical forecasting, hallucinating diverse but incorrect patterns could yield better FID scores than accurate prediction. Our setup aligns with domain standards: HyperDM [1] and Price et al. [2] rely on RMSE, CRPS, and SSIM rather than FID. Thus, we prioritize MSE and SSIM to demonstrate accurate tracking of the conditional target.
> - AQ1: We agree that restoration and style transfer are interesting applications. However, as discussed in **W3**, our work specifically targets **safety-critical predictive tasks** where the goal is fidelity to a ground truth rather than generating diverse artistic outputs. We believe our experiments on reconstruction and real-world weather forecasting provide a rigorous evaluation of the method's utility in this targeted domain. We leave the extension to stochastic generation tasks like style transfer for future work.
> - AQ2: The contrastive component in our method follows the core principle of contrastive learning, encouraging similarity between positives and dissimilarity from negatives, without relying on view-based augmentations. While many recent approaches construct positives through transformations of the same input, this is only one successful strategy, not a requirement of the framework. In our formulation, the supervised FM target velocity $(x_1 − x_0)$ serves as the anchor, the model’s predicted velocity under an ID conditioning provides the positive sample, and the prediction under a pseudo-OOD conditioning acts as the negative one. The triplet-margin loss, therefore, enforces the standard contrastive geometry directly in the vector-field space. We will clarify this to avoid implying that data augmentations are mandatory for contrastive learning.
> - AQ3 and minor comments: We will add the full weather forecasting experiment settings to the appendix. We will also revise the manuscript to incorporate the corrections noted in the minor comments.
>
> [1] Chan et al. (2024). Estimating epistemic and aleatoric uncertainty with a single model. Advances in Neural Information Processing Systems, 37.
>
> [2] Price et al. (2025). Probabilistic weather forecasting with machine learning. Nature.

---

### Meta-Review · Area_Chair_Zh77 · 2026-01-08

**Summary:**

This paper primarily has negative ratings from reviewers mainly because of poor experimental results. Reviewers (DmnZ, jvGi) questioned the "straightness" assumption of inference trajectories, the specific definition of the "context" variable, and the omission of FID scores in favor of reconstruction metrics. Some reviewers also raised concerns regarding the "in-distribution only" training setup and the potential for "unfair" baseline comparisons against DiffPath, and the statistical guarantees of the conformal prediction application. Reviewer CdUW expressed skepticism over the "near-perfect" AUROC results, suggesting the benchmarks might be too trivial.

I think the paper has very weak experimental section. For image datasets, authors consider very simple datasets such as SVHN or MNIST, while there are so many harder datasets. For a top-tier conference like ICLR, the experimental results should be very strong, and more ablations/analysis needs to be performed. Due to these reasons, I vote for rejecting the paper, and I encourage the authors to work on these for their next version.

**Reviewer Concerns:**

In the rebuttal, authors clarified some confusions/concerns that reviewers had - that the OOD score measures velocity field stability rather than path straightness, provided a rigorous definition of the context variable, and justified the use of SSIM/MSE over FID for safety-critical forecasting. However, the main issues regarding experimental results still remain.

**Reviewer Scores:**

Reviewers would most likely retain their scores. While some of the misunderstandings were clarified, the paper still has weak experimental section.

---

### Decision · Program_Chairs · 2026-01-26

Reject